# Mental health and illness of medical students and newly graduated doctors during the pandemic of SARS-Cov-2/COVID-19

**Lis Campos Ferreira** [1,2]* , **Rívia Siqueira Amorim**[3‡], **Fellipe Matos Melo Campos**[3‡], **Rosana Cipolotti**[1]

**1** Postgraduate Program in Health Sciences, Federal University of Sergipe, Aracaju, Sergipe, Brazil, **2** Department of Medicine, Tiradentes University, Aracaju, Sergipe, Brazil, **3** Department of Medicine, Federal University of Sergipe, Aracaju, Sergipe, Brazil

☉ These authors contributed equally to this work.
‡ These authors also contributed equally to this work.
* liscamposf@yahoo.com.br

**Data Availability Statement:** All relevant data are within the manuscript and its Supporting information files.

## Abstract

Introduction: SARS-Cov-2 virus pandemic causes serious emotional consequences. It has occurred widespread medical courses suspension, and graduations were anticipated. Field hospitals, set up to treat patients with mild to moderate COVID-19, were the main work-places of newly graduated doctors. Objective: To assess the impact of SARS-Cov-2/COVID-19 pandemic on mental health of medical interns and newly graduated doctors. Method: This is a cross-sectional study performed using a digital platform. Links to forms were sent in two moments: moment 1 (M1), at the beginning of the pandemic, in the first half of April/2020 and moment 2 (M2), after six months of pandemic, in the second half of September/2020. All students from the medical internship and all doctors graduated since 2018 from the three medical schools in Sergipe-NE-Brazil were invited. Results: 335 forms were answered in April and 148 in September. In M1 88.9% considered themselves exposed to excess of information about COVID-19, which was associated with anxiety symptoms (p = 0.04). Long family physical distance was also associated with these symptoms, as increased appetite (p = 0.01), feeling shortness of breath (p = 0.003) and sweating (p = 0.007). Fear of acquire COVID-19 was reported as intense by almost half of participants, and of transmitting by 85.7% in M1. In M2 41.2% reported the death of friends or relatives. Psychiatric illness was described by 38.5% and psychotropic drugs use by 30.1% in M1, especially those who lived alone (p = 0.03) and the single ones (p = 0.01). Alcohol intake was reported by 54.3%, and among doctors graduated in 2020 it increased from 50% in M1 to 85% in M2 (p = 0.04). Conclusion: The pandemic had a negative impact on the mental health of medical students and newly graduated doctors. Exposure to excessive COVID-19 information and family physical distance were associated to anxiety symptoms. Among doctors graduated in 2020, alcohol intake increased during pandemic evolution.

**Funding:** The author Rosana Cipolotti received funding from Postgraduate Support Program (PROAP) - CAPES - Ministry of Education - a total of R$ 4000,00 (four thousand reais). The funders had no role in study design, data collection and analysis, decision to publish, or preparation of the manuscript.

**Competing interests:** The authors have declared that no competing interests exist.

## Introduction

The SARS-Cov-2 virus pandemic, which causes COVID-19, is resulting in serious emotional consequences for the world population. As countries struggle to manage the waves of physical illness and death, there is evidence that a new wave is taking shape due to the increase in mental disorders and substance abuse [1].

Healthcare workers are particularly exposed to the risk of becoming infected and contaminating people in their personal environment, which reflects in greater emotional damage. In a survey involving 2182 individuals, it was observed that doctors had a higher frequency of sleep disorders, anxiety, depression, somatization and obsessive-compulsive symptoms than non-medical health professionals [2]. Compared to the administrative team, frontline doctors in the Emergency and Intensive Care Units were 1.4 times more likely to be afraid and twice as likely to experience anxiety and depression [3].

In Brazil, once a health emergency has been decreed, medical courses were suddenly interrupted, and undergraduate students were later directed to activities, when not totally remote, limited to a few scenarios of face-to-face practice [4]. Field hospitals, set up to treat and isolate patients with mild to moderate COVID-19, were the workplaces of newly graduated doctors, who reduced labor shortages especially during peak infection [5, 6]. For this, graduations were anticipated and the newly graduated doctors entered the emergency shifts of flu-like syndromes [7].

Mental stress represents the main environmental risk factor for psychiatric illnesses and a state of prolonged sustained stress can increase the propensity to depression and other mental disorders [8]. Investigating the mental health situation of medical interns and newly graduated doctors is of great importance in the planning and execution of strategies to prevent and deal with potential injuries and, consequently, better performance of professional activity. Thus, the aim of the present study was to assess the impact of the COVID-19 pandemic on the mental health of these individuals in Brazil.

## Materials and methods

This is a cross-sectional descriptive study carried out during the first half of April (moment 1-M1—one month after the beginning of the COVID-19 pandemic) and second half of September 2020 (moment 2—M2—six months after) using a form on the GoogleForms® digital platform. The project was approved by the ethical committee of Federal University of Sergipe represented by the number 4.046.521, and digital consent form was obtained from all participants. All medical interns (fifth and sixth years of medical school) and doctors graduated since 2018 in the three medical schools (one private and two publics) in the state of Sergipe, located in the northeast of Brazil, were invited to participate by e-mail obtained from the university database. The invitations were sent, at both times of the survey, to the same registered e-mails and the link was available for fifteen days.

The inclusion criteria were: i) to be attending a medical internship or ii) to be a doctor graduated in 2018, 2019 or 2020, attending or not attending a medical residency and, in both cases, to be over 18 years old. Incomplete, blank or repeated forms were excluded.

The questionnaires were anonymous, guaranteeing data privacy and confidentiality. They consisted of closed-answer questions (multiple-choice, single-answer, dichotomous-answer), matrix (Likert scale), and open-answer questions. In order to identify the participants who had answered both questionnaires, it was asked in M2 if the participant had answered the form in M1.

The same sociodemographic data were collected in M1 and M2 to characterize the population, including age, sex, relationship status, housing conditions and cohabitants. It was also

inquired about general medical conditions, as COVID-19 infection, H1N1 vaccine and comorbidities. At the time of data collection there were no vaccine available for COVID-19. Regarding psychosocial assessment, data about sources of COVID-19 information, fear of COVID-19 infection and financial loss, presence of anxiety symptoms, support from colleagues and university and strategies to reduce distance from family and friends were collected in M1. In order to draw a parallel between fears in M1 and reality in M2, questions about COVID-19 outcomes in participants and their loved ones were added in M2. Mental health history was asked in M1 and M2, including psychiatric diagnosis, use of psychotropic drugs, and legal or illegal psychoactive substances. In M2, changes in the pattern of alcohol intake, use of tobacco or illicit drugs during the pandemic were added, in addition to diagnosis of new psychiatric illness or change in current psychiatric treatments. Additional information about the M1 and M2 questionnaires is available at S1 File.

The population size of the study was 1000 individuals, and all of them were invited to participate by email. It was included everyone who answered the web survey and met the selection criteria. The data were extracted in an Excel® table, and simple averages and frequencies were calculated for descriptive analysis. For comparative analysis, the Epi Info® version 7 statistical program and two-tailed chi-square tests were used to analyze categorical and proportional variables, and the paired t-test for continuous variables. P was considered statistically significant if less than 0.05.

## Results

In M1, 394 responses were obtained. After excluding repeated and incomplete forms, 335 were included for data analysis (response rate of 33.5%). In M2, 169 responses were obtained and, after applying the exclusion criteria, resulted in 148 forms (response rate of 14.8%). For descriptive analysis of this second moment, all 148 forms were considered. Finally, for comparative analysis in M1 and M2, 14 forms from participants who reported not having answered the questionnaire in M1 were excluded, resulting in 134 respondents.

General characteristics of the participants and sociodemographic data were similar in M1 and M2, as described in Table 1.

In M1, 104 participants (31%) did not consider themselves to have sufficient information about COVID-19 and 127 (38%) reported seeking information frequently. The sources of information on COVID-19 most used by the research participants were Ministry of Health / World Health Organization (89.2%), scientific journals (73.4%), social media (40.9%) and television (36.7%).

Nevertheless, 88.9% found themselves exposed to an excessive amount of COVID-19 information, which was associated to the presence of anxiety symptoms (p = 0.04). Long family physical distance, represented by living in a different state, was associated with some of these symptoms, including increased appetite (p = 0.01), feeling of shortness of breath (p = 0.003) and sweating (p = 0.007), as shown in Fig 1.

In M1, almost half of the participants (n = 161) reported intense fear of having COVID-19, 287 (85.7%) of transmitting the virus and 87.8% of dying from COVID-19. Another fear, related to economic loss during the pandemic, was referred by 212 (63.3%) participants in M1. In M2 this fear was converted into a real financial loss for 38 (25.6%) of them. In M1 none of the participants had confirmed COVID-19 diagnosis, while in M2 15% of them had laboratory confirmation of the infection, one of whom required hospitalization. When asked about death of their loved ones by COVID-19, 61 (41.2%) reported knowing at least one person who had died from the disease in M2.

**Table 1. General characteristics of the participants in the two moments of the research.**

|  | April/2020 (M1)* (n = 335) | September/2020 (M2)* (n = 148) | *P* value |
|---|---|---|---|
| **Sex** |  |  | 0.41 |
| **Women** | 198 (59.1%) | 94 (63.5%) |  |
| **Men** | 137 (40.9%) | 54 (36.5%) |  |
| **Mean age (years)** | 25.6 | 26.2 | 0.26 |
| **Relationship status** |  |  | 1 |
| **Married / Stable union** | 35 (10.5%) | 21 (14.2%) |  |
| **Dating / Engaged** | 126 (37.6%) | 50 (33.8%) |  |
| **Single / Widow(er) / Divorced** | 174 (51.9%) | 77 (52%) |  |
| **Cohabitants** |  |  | 0.22 |
| **Partner** | 41 (12.2%) | 22 (14.9%) |  |
| **Relatives** | 201 (60%) | 91 (61.5%) |  |
| **Colleagues** | 38 (11.3%) | 18 (12.2%) |  |
| **None** | 62 (18.5%) | 20 (13.5%) |  |
| **University training internship** |  |  | 0.91 |
| **9th semester** | 67 (20%) | 39 (26.3%) |  |
| **10th semester** | 59 (17.6%) | 25 (16.9%) |  |
| **11th semester** | 56 (16.7%) | 22 (14.9%) |  |
| **12th semester** | 34 (10.2%) | 8 (5.4%) |  |
| **Graduated attending medical residency** | 43 (12.8%) | 13 (8.8%) |  |
| **Graduated without attending medical residency** | 76 (22.7%) | 41 (27.7%) |  |
| **University** |  |  | 0.37 |
| **Public–*Campus* 1** | 95 (28.4%) | 41 (27.7%) |  |
| **Public–*Campus* 2** | 125 (37.3%) | 63 (42.6%) |  |
| **Private** | 115 (34.3%) | 44 (29.7%) |  |

*M1: Moment 1; M2: Moment 2.

About habits and addictions, in M1 182 participants (54.3%) reported alcohol intake, 12 reported smoking and 18 (5.4%) use of illicit drugs (13 marijuana, two ecstasy, one LSD, one cocaine and three did not detail). In M2, six users increased the frequency of illicit drug use and one individual started using it.

In relation to psychiatric history, in M1 129 participants (38.5%) reported diagnosed psychiatric illness, among anxiety disorder (n = 105), major depressive disorder (n = 47), bipolar disorder (n = 6), attention deficit hyperactivity disorder (ADHD) (n = 5), eating disorder (n = 4), post-traumatic stress disorder (PTSD) (n = 3), obsessive-compulsive disorder (OCD) (n = 3), adjustment disorder (n = 1) and chemical dependency (n = 1). In M2, new psychiatric diagnoses were found in 15 participants, among burnout (n = 2), anxiety disorder (n = 7), depressive disorder (n = 4), ADHD (n = 1) and OCD (n = 1). Women had more psychiatric comorbidities (p = 0.001). Living alone was associated with higher rate of depression (p = 0.000001).

Regarding psychiatric treatment, a total of 101 participants (30.1%) used psychotropic medication in M1, especially those who lived alone (p = 0.03) and the single ones (p = 0.01). The use of psychotropic medication was more frequent in M1 compared to M2 (p = 0.009), when among 21.6% (n = 32) of the participants who used it, 13 remained at the same prescribed dose, eight increased the dose, and eleven started using it. The most frequently cited medications in M2 were antidepressants (n = 25), non-benzodiazepine sleep inducers (n = 8),

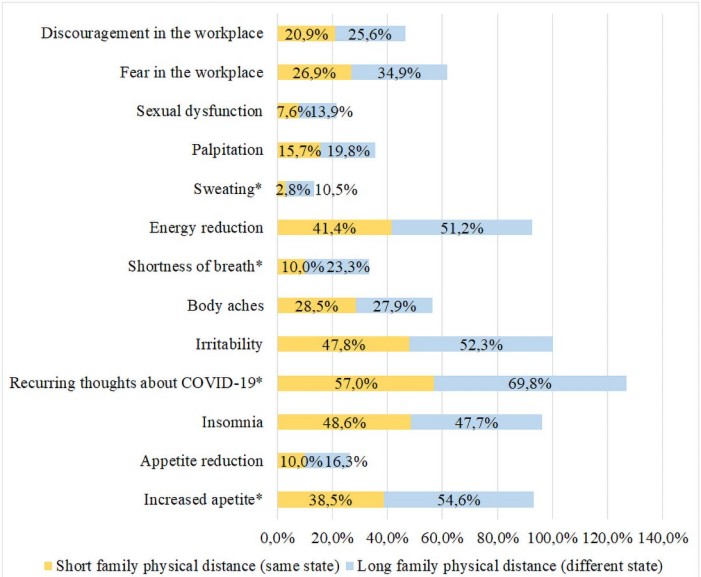

**Fig 1. Distribution of anxiety symptoms according to the family physical distance.** Symptoms with statistical relevance (p <0.05) are represented by *.

benzodiazepines (n = 3), mood stabilizers (n = 5), antipsychotics (n = 2) and psychostimulants (n = 2).

Nineteen participants in M2 reported to be working directly in the care of patients with COVID-19 (12.8%). Of these, 14 (73.6%) started or increased the use of psychotropic drugs in the period (p = 0.04). There was no difference regarding the diagnosis of mental illness, use of psychotropic drugs or illicit drugs between interns and doctors. The data on mental health are shown in Table 2.

Evaluating particularly the group of doctors graduated in 2020, alcohol intake increased from 50% in M1 to 85% in M2 (p = 0.04). Information on the diagnosis of psychiatric illness showed no difference between the two moments of the study (p = 0.64), as well as information on the use of psychotropic drugs (p = 0.61), as shown in Table 3.

## Discussion

Cyberchondria, which means excessive Internet search on health-related issues, is a common behavior, but it can lead to high levels of concern and anxiety [9]. In the present study, 88.9% of the participants considered themselves exposed to an excess of information about COVID-19. Besides, a higher frequency of anxiety symptoms was observed in these same patients (p = 0.04), in agreement with previous study in which cyberchondria showed positive correlations with virus anxiety during the pandemic [10].

Nevertheless, 31% of participants in this study reported that they were not well informed about COVID-19, which reinforces the need for good quality information, through certified and reliable sources [11]. Although approximately 90% of the participants sought official sources from the Ministry of Health / World Health Organization, 41% also used social media, which can lead to high exposure to low quality information.

Another factor associated with the presence of anxiety symptoms was the long family physical distance, since they were living in a different state. When isolation measures were implemented, due to the great territorial extension of Brazil, living with other people may have had

**Table 2. Factors associated with the participants' mental health at both times of the research.**

| | April/2020 (M1)* (n = 335) | September/2020 (M2)* (n = 134) | P value |
|---|---|---|---|
| **Alcohol intake** | | | 0.0004 |
| No | 153 (45.7%) | 37 (27.6%) | |
| Yes | 182 (54.3%) | 97 (72.4%) | |
| **Smoking habit** | | | 0.84 |
| No | 323 (96.4%) | 128 (95.5%) | |
| Yes | 12 (3.6%) | 6 (4.5%) | |
| **Use of illicit drugs** | | | 0.04 |
| No | 317 (94.6%) | 119 (88.8%) | |
| Yes | 18 (5.4%) | 15 (11.2%) | |
| **Psychiatric disorder** | | | 0.91 |
| - No | 206 (61.5%) | 81 (60.4%) | |
| - Yes | 129 (38.5%) | 53 (39.6%) | |
| Anxiety | 105 (31.3%) | 41 (30.6%) | 0.96 |
| Depression | 47 (14%) | 16 (11.9%) | 0.65 |
| Bipolar | 6 (1.8%) | 2 (1.5%) | 1 |
| OCD | 3 (0.9%) | 4 (3%) | 0.1 |
| ADHD | 5 (1.5%) | 5 (3.7%) | 0.15 |
| Eating disorder | 4 (1.2%) | 6 (4.5%) | 0.03 |
| PTSD | 3 (0.9%) | 1 (0.7%) | 1 |
| Adjustment disorder | 1 (0.3%) | 0 | 1 |
| Chemical dependency | 1 (0.3%) | 0 | 1 |
| Burnout | 0 | 1 (0.7%) | 0.28 |
| **Psychotropic treatment** | | | 0.009 |
| - No | 234 (69.9%) | 110 (82.1%) | |
| - Yes | 101 (30.1%) | 24 (17.9%) | |

*M1: Moment 1; M2: Moment 2; OCD: Obsessive-compulsive disorder; ADHD: Attention deficit hyperactivity disorder; PTSD: Post-traumatic stress disorder.

a protective effect on mental health compared to those who lived alone. In agreement with this finding, a previous study in Indonesia assessed the effect of physical distance on the level of anxiety and found moderate to severe levels of anxiety in 40.3% of the sample [12].

The anxiety that arises in unpredictable situations such as a pandemic, and the fear of the unknown can underlie the fear related to SARS-Cov-2 [13, 14]. As many infected people are asymptomatic, they are not diagnosed in time for isolation during the viral transmission phase. In addition, incidence and mortality become inaccurate [13]. This probably explains the intense fear of falling ill reported by half of the participants in M1, and that of transmitting by 85.7% of them. There is a greater fear of transmitting the virus than being infected by it, which corroborates the study by Mertens et al (2020), in which the risk of illness of loved ones was considered a predictor for the fear of COVID-19 [14].

In M2, 41.2% reported death by COVID-19 of friends or relatives. The isolation of suspected and infected patients, physical contact with loved ones replaced by remote audiovisual connections, restricted visits and the absence of funeral rituals made the grieving process more difficult and emotionally exhausting [15].

In addition to all personal and family losses, the fear of unfavorable financial outcomes was also present in 63.3% of people in M1. However, six months later (M2), the percentage of people whose financial situation really worsened was reduced to 25.6%. This finding possibly

**Table 3. Factors associated with the mental health of doctors graduated in 2020 at both times of the research.**

| | April/2020 M1* (n = 12) | September/2020 M2* (n = 20) | P value |
|---|---|---|---|
| **Alcohol intake** | | | 0.04 |
| **No** | 6 (50%) | 3 (15%) | |
| **Yes** | 6 (50%) | 17 (85%) | |
| **Smoking habit** | | | NA |
| **No** | 12 (100%) | 20 (100%) | |
| **Yes** | 0 | 0 | |
| **Use of illicit drugs** | | | 0.51 |
| **No** | 10 (83.3%) | 18 (90%) | |
| **Yes** | 2 (16.7%) | 2 (10%) | |
| **Psychiatric disorder** | | | 0.64 |
| - **No** | 9 (75%) | 17 (85%) | |
| - **Yes** | 3 (25%) | 3 (15%) | |
| **Anxiety** | 2 (16.7%) | 3 (15%) | |
| **Depression** | 1 (8.3%) | 1 (5%) | |
| **Bipolar** | 0 | 0 | |
| **OCD** | 0 | 0 | |
| **ADHD** | 0 | 1 (5%) | |
| **Eating disorder** | 0 | 0 | |
| **PTSD** | 0 | 0 | |
| **Adjustment disorder** | 0 | 0 | |
| **Chemical dependency** | 0 | 0 | |
| **Burnout** | 0 | 0 | |
| **Psychotropic treatment** | | | 0.61 |
| - **No** | 10 (83.3%) | 18 (90%) | |
| - **Yes** | 2 (16.7%) | 2 (10%) | |

*M1: Moment 1; M2: Moment 2; OCD: Obsessive-compulsive disorder; ADHD: Attention deficit hyperactivity disorder; PTSD: Post-traumatic stress disorder; NA: Non-available.

reflects personal and family adaptations, with reduced costs. Besides, the Ministry of Health launched the program "Brazil counts on me—Health Professionals", which recruited doctors and other health professionals, including newly graduated ones, to work in COVID-19 patients care, with financial gain due to the increased availability of shifts [16].

The co-occurrence of anxiety and substance use disorders has been well established in the literature. There is also a social acceptability of using alcohol as a "stress reliever" among higher income and educational groups [17]. In M1, alcohol intake was identified in 54.3% of the participants in the present study. In Poland, it was also observed a significant increase in alcohol consumption by doctors during the quarantine [18]. Evaluating particularly the group of doctors graduated in 2020, alcohol intake increased from 50% in M1 to 85% in M2 (p = 0.04). It is important to note that this specific group faced unprecedented challenges related to the pandemic. The end of the internship was modified, with the suspension of some activities, loss of space for practice scenarios, in addition to sudden unscheduled migration to remote classes, with students and teachers in challenging adaptation to new teaching strategies [19]. There was also a loss of graduation rite, cancellation of graduation ceremonies and parties, due to measures of social distance. Finally, early graduation allowed them to enter a highly demanding workplace (field hospitals and emergency rooms) with little preparation and experience [20].

M1 data described diagnosed psychiatric illness in 38.5% of the sample. Anxiety disorder (81.4%) and major depressive disorder (36.4%) were the main diagnoses, and more frequent in women (p = 0.001), an association already evidenced in previous studies [10, 18, 21, 22]. In M2 a new diagnosis of psychiatric comorbidity was observed in 10% of the sample, with an increase in the number of participants who reported eating disorders and obsessive-compulsive disorder, which suggests that anxiety in the face of pandemic uncertainties may have precipitated compulsive behaviors [23].

In M1, 30% of the participants used psychotropic medication, especially those who lived alone (p = 0.03) and the single ones (p = 0.01), as reported in previous study [24]. There have been few reported cases of self-prescription, which suggests that most participants had adequate access to mental health specialists, either in person or by telemedicine. Another study concluded that, throughout the pandemic, many psychiatrists started to do remote care, which may have expanded the possibilities of access [25]. Psychotropic treatment increased (p = 0.03) among physicians who worked directly in the COVID-19 patients care, possibly due to the greater stress faced during the pandemic [26, 27].

The present study has as main limitations the low response rates, which is expected in web surveys compared to other data collection modes [28, 29], the reduction of the sample size in M2 in relation to M1, the impossibility of matching responses per person due to the mandatory anonymity, and the existence of questions asked only in M1 or in M2, due to the nexus relationship with the epidemiological moment.

## Conclusions

In conclusion, SARS-CoV-2 pandemic had a negative impact on the mental health of medical students and newly graduated doctors. Exposure to excessive COVID-19 information and family physical distance were associated to anxiety symptoms. Working directly with COVID-19 patients was associated to higher use of psychotropic drugs. Among doctors graduated in 2020, the alcohol intake increased in the 6-month period of evaluation.

## Supporting information

**S1 File.**
(DOCX)

**S1 Database.**
(XLSX)

## Author Contributions

**Conceptualization:** Lis Campos Ferreira, Rívia Siqueira Amorim, Fellipe Matos Melo Campos, Rosana Cipolotti.

**Data curation:** Lis Campos Ferreira, Rívia Siqueira Amorim, Fellipe Matos Melo Campos, Rosana Cipolotti.

**Formal analysis:** Lis Campos Ferreira, Rosana Cipolotti.

**Funding acquisition:** Rosana Cipolotti.

**Investigation:** Lis Campos Ferreira, Rívia Siqueira Amorim, Fellipe Matos Melo Campos, Rosana Cipolotti.

**Methodology:** Lis Campos Ferreira, Rívia Siqueira Amorim, Fellipe Matos Melo Campos, Rosana Cipolotti.

**Writing – original draft:** Lis Campos Ferreira.

**Writing – review & editing:** Rívia Siqueira Amorim, Fellipe Matos Melo Campos, Rosana Cipolotti.

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
