## [Decision Letter · Decision Letter 0]

12 Feb 2021

PONE-D-20-40515

Mental health and illness of medical students and newly graduated doctors during the pandemic of SARS-Cov-2/COVID-19

PLOS ONE

Dear Dr. Ferreira,

Thank you for submitting your manuscript to PLOS ONE. After careful consideration, we feel that it has merit but does not fully meet PLOS ONE’s publication criteria as it currently stands. Therefore, we invite you to submit a revised version of the manuscript that addresses the points raised during the review process.

We look forward to receiving your revised manuscript.

Kind regards,

Geilson Lima Santana, M.D., Ph.D.

Academic Editor

PLOS ONE

Journal Requirements:

3. Please provide a sample size and power calculation in the Methods, or discuss the reasons for not performing one before study initiation.

Reviewers' comments:

Reviewer's Responses to Questions

**Comments to the Author**

1. Is the manuscript technically sound, and do the data support the conclusions?

Reviewer #1: Partly

Reviewer #2: Partly

2. Has the statistical analysis been performed appropriately and rigorously? 

Reviewer #1: Yes

Reviewer #2: Yes

3. Have the authors made all data underlying the findings in their manuscript fully available?

Reviewer #1: Yes

Reviewer #2: Yes

4. Is the manuscript presented in an intelligible fashion and written in standard English?

Reviewer #1: Yes

Reviewer #2: No

5. Review Comments to the Author

Reviewer #1: One of the big limitations is that whether the participants at M1 and M2 were the same individuals, also not know that previous condition on the participants pre-the pandemic.

Another is small sample, thus, the results may not represent the actual status.

therefore, the conclusion should be revised.

Reviewer #2: The paper findings are truly interesting, but the presented work requires revision before publication. In general, the paper brings a great amount of information regarding the stated objective: assess the impact of COVID-19 pandemics on the mental health of students and recently graduated doctors. Nonetheless, the paper also lacks organization when presenting their findings and reports data that was not explained in their methods and is not clearly related to their aim, such as comparing subjects that lived with or without their families. The paper has great potential, but should be clearer and focus on its objective and findings. Although a research project may be very extensive, a paper should be able to summarize findings and report them in a comprehensive way to the scientific community.

Authors should also check their English.

ABSTRACT:

The authors should consider including more specific information about COVID-19 impacts on mental health in their introduction.

INTRODUCTION:

Literature Review was well done. Objective and research justification are very clear.

METHODS:

The authors should better explain why they considered the study a longitudinal cohort. Most of the data used to extract their conclusions were based on single time point analysis (transversal analysis).

The authors should state in the methods that the work was approved by the ethical committee.

The authors should consider a better way to report the questions made in M1 and M2 and also to make it clearer which questions were asked at both time-points.

It would be of great benefit to explain which variables were used to characterize the population and each one were your dependent variables to analyze mental health and, if possible, why.

RESULTS:

Data is only presented in tables, which makes it harder to visualize informations. The authors should consider using charts to illustrate their main findings.

The authors should avoid making data interpretation in the results.

The authors should consider making it clearer what each data reported is about: population baseline, characteristics, mental health evaluation across time, etc

DISCUSSION:

Although the discussion is well-argued, it lacks connection between paragraphs, and sometimes comprehension is undermined. The authors should consider rearranging the sequence or adding subtitles to this section.

Conclusion:

The authors should be more specific on their conclusion in line 340

6. PLOS authors have the option to publish the peer review history of their article (what does this mean?). If published, this will include your full peer review and any attached files.

Reviewer #1: No

Reviewer #2: **Yes: **Lucas Albuquerque Chinelatto

---

## [Author Response · Author response to Decision Letter 0]

19 Mar 2021

Authors’ Response to the Review Comments

Journal: PLOS ONE

Manuscript: PONE-D-20-40515

Title of Paper: Mental health and illness of medical students and newly graduated doctors during the pandemic of SARS-Cov-2/COVID-19

Dear Editor and Reviewers,

We appreciate the efforts in carefully reviewing this manuscript in this difficult time. The authors agreed with all comments, and we made adjustments and corrections. We are grateful for the suggestions that improved the quality of our article, and believe that the revised version can meet the jornal publication requirements.

Editor Comments:

 Corrections were made in order to meet PLOS ONE's style requirements.

 A copy of the questionnaires developed as part of this study was included as Supporting Information. 

3. Please provide a sample size and power calculation in the Methods, or discuss the reasons for not performing one before study initiation.

 It was included in methods that the sample size and the population size were the same: “The population size of the study was 1000 individuals, and all of them were invited to participate by email. It was included everyone who answered the web survey and met the selection criteria.”

 The information was rectfied. We declare that the author Rosana Cipolotti received funding from Postgraduate Support Program (PROAP) - CAPES - Ministry of Education - a total of R$ 4000,00 (four thousand reais). The funders had no role in study design, data collection and analysis, decision to publish, or preparation of the manuscript.

 We completely agree. All relevant data are within the manuscript and its Supporting Information files.

Review Comments:

Reviewer #1: One of the big limitations is that whether the participants at M1 and M2 were the same individuals, also not know that previous condition on the participants pre-the pandemic. Another is small sample, thus, the results may not represent the actual status. Therefore, the conclusion should be revised.

 Thank you for your important comment. It was explained in methods that in order to identify the participants who had answered both questionnaires, it was asked in M2 if the participant had answered the form in M1. Pre-pandemic psychiatric condition of the participants was also accessed, and conclusions were revised.

Reviewer #2: The paper findings are truly interesting, but the presented work requires revision before publication. In general, the paper brings a great amount of information regarding the stated objective: assess the impact of COVID-19 pandemics on the mental health of students and recently graduated doctors. Nonetheless, the paper also lacks organization when presenting their findings and reports data that was not explained in their methods and is not clearly related to their aim, such as comparing subjects that lived with or without their families. The paper has great potential, but should be clearer and focus on its objective and findings. Although a research project may be very extensive, a paper should be able to summarize findings and report them in a comprehensive way to the scientific community. Authors should also check their English.

ABSTRACT:

The authors should consider including more specific information about COVID-19 impacts on mental health in their introduction.

INTRODUCTION:

Literature Review was well done. Objective and research justification are very clear.

METHODS:

The authors should better explain why they considered the study a longitudinal cohort. Most of the data used to extract their conclusions were based on single time point analysis (transversal analysis). The authors should state in the methods that the work was approved by the ethical committee. The authors should consider a better way to report the questions made in M1 and M2 and also to make it clearer which questions were asked at both time-points. It would be of great benefit to explain which variables were used to characterize the population and each one were your dependent variables to analyze mental health and, if possible, why.

RESULTS:

Data is only presented in tables, which makes it harder to visualize informations. The authors should consider using charts to illustrate their main findings.

The authors should avoid making data interpretation in the results.

The authors should consider making it clearer what each data reported is about: population baseline, characteristics, mental health evaluation across time, etc

DISCUSSION:

Although the discussion is well-argued, it lacks connection between paragraphs, and sometimes comprehension is undermined. The authors should consider rearranging the sequence or adding subtitles to this section.

CONCLUSION:

The authors should be more specific on their conclusion in line 340

 We fully agree with the comments. Information about COVID-19 impacts on mental health was added to the abstract. In methods, after discussion among the authors, we understood that this is a cross-sectional descriptive study. We also added the approval number on the ethics committee. Variables were described, and the whole methodology has been rewritten. A figure was added to the article to facilitate understanding of the data. Results were rewritten, discussion was rearranged to provide better comprehension, and conclusions were revised.

---

## [Decision Letter · Decision Letter 1]

13 Apr 2021

PONE-D-20-40515R1

Mental health and illness of medical students and newly graduated doctors during the pandemic of SARS-Cov-2/COVID-19

PLOS ONE

Dear Dr. Ferreira,

Thank you for submitting your manuscript to PLOS ONE. After careful consideration, we feel that it has merit but does not fully meet PLOS ONE’s publication criteria as it currently stands. Therefore, we invite you to submit a revised version of the manuscript that addresses the points raised during the review process.

We look forward to receiving your revised manuscript.

Kind regards,

Geilson Lima Santana, M.D., Ph.D.

Academic Editor

PLOS ONE

Journal Requirements:

Reviewers' comments:

Reviewer's Responses to Questions

**Comments to the Author**

1. If the authors have adequately addressed your comments raised in a previous round of review and you feel that this manuscript is now acceptable for publication, you may indicate that here to bypass the “Comments to the Author” section, enter your conflict of interest statement in the “Confidential to Editor” section, and submit your "Accept" recommendation.

Reviewer #2: All comments have been addressed

2. Is the manuscript technically sound, and do the data support the conclusions?

Reviewer #2: Yes

3. Has the statistical analysis been performed appropriately and rigorously? 

Reviewer #2: Yes

4. Have the authors made all data underlying the findings in their manuscript fully available?

Reviewer #2: Yes

5. Is the manuscript presented in an intelligible fashion and written in standard English?

Reviewer #2: Yes

6. Review Comments to the Author

Reviewer #2: Authors have solved most issues. Even so, there are minor issues that must be addressed before publication:

- Authors should consider to make it clearer that they are writing specifically about Brazil from line 60 and beyond.

- Figure 1 is very interesting, but authors should consider using percentages (%) instead of absolute values, as the figure does not make it clear if the association is positive or negative. Characteristics that had statistically relevant association appear to have similar absolute numbers in terms of quantity. One can infer that this is because the "long family physical distance" group has a smaller n size, and therefore a smaller absolute value distance represents a higher incidence in this group (increased %). Authors should seek to make this interpretation clearer and independent from text reading.

- Authors should include a Figure 1 legend to include "*" meaning.

- Please, check english in line 127-128

- Line 272-273 statement is not clear. What is the correlation between some doctors self-prescribing and an adequate access to mental health?

7. PLOS authors have the option to publish the peer review history of their article (what does this mean?). If published, this will include your full peer review and any attached files.

Reviewer #2: No

---

## [Author Response · Author response to Decision Letter 1]

14 Apr 2021

Reviewer #2: 

Authors should consider to make it clearer that they are writing specifically about Brazil from line 60 and beyond. 

We agree and have included the missing information.

Figure 1 is very interesting, but authors should consider using percentages (%) instead of absolute values, as the figure does not make it clear if the association is positive or negative. Characteristics that had statistically relevant association appear to have similar absolute numbers in terms of quantity. One can infer that this is because the “long readin physical distance” group has a smaller n size, and therefore a smaller absolute value distance reading ent a higher incidence in this group (increased %). Authors should seek to make this interpretation clearer and reading ente from text reading. 

We changed the figure as suggested.

Authors should include a Figure 1 legend to include "*" meaning. 

Legend was included.

Please, check english in line 127-128. 

Phrase has been reworded.

Line 272-273 statement is not clear. What is the correlation between some doctors self-prescribing and an adequate access to mental health?

The word that should have been used was "few", not "some". We wanted to say that as only a few doctors prescribed themselves, we can infer that in the rest of the cases there was assistance from a mental health specialist, since a significant percentage of the participants was using psychotropic drugs.

---

## [Editor Report · Decision Letter 2]

28 Apr 2021

Mental health and illness of medical students and newly graduated doctors during the pandemic of SARS-Cov-2/COVID-19

PONE-D-20-40515R2

Dear Dr. Ferreira,

We’re pleased to inform you that your manuscript has been judged scientifically suitable for publication and will be formally accepted for publication once it meets all outstanding technical requirements.

Kind regards,

Geilson Lima Santana, M.D., Ph.D.

Academic Editor

PLOS ONE
---

## [Editor Report · Acceptance letter]

3 May 2021

PONE-D-20-40515R2 

Mental health and illness of medical students and newly graduated doctors during the pandemic of SARS-Cov-2/COVID-19 

Dear Dr. Ferreira:

I'm pleased to inform you that your manuscript has been deemed suitable for publication in PLOS ONE. Congratulations! Your manuscript is now with our production department. 

Kind regards, 

on behalf of

Dr. Geilson Lima Santana 

Academic Editor

PLOS ONE